# Numerical Models for Simulating Ocean Physics

Michael J. Bell[1], Andreas Schiller[2], Stefania Ciliberti[3]

[1]MetOffice, Exeter, UK
[2]CSIRO Environment, Castray Esplanade, Hobart, Tasmania, Australia
[3]Nologin Oceanic Weather Systems, Santiago de Compostela, Spain

*Correspondence to*: Mike Bell (mike.bell@metoffice.gov.uk)

**Abstract.** We describe, at an elementary level, the spatially varying properties of the ocean that physical ocean models represent, the principles they use to evolve these properties with time, the physical phenomena that they simulate, and some of the roles these phenomena play within the Earth system. We describe at an intermediate level the governing equations the models use and the grids that they typically use, and at a more advanced technical level, the methods and approximations that the models use and the difficulties that limit their accuracy or reliability. We also briefly describe the wider context and future prospects for the development of these models.

## 1 Introduction

The models of ocean physics described in this paper, use physical principles to simulate how the three-dimensional structures of the ocean's temperature, salinity and currents evolve in time. Section 2 describes the models at an introductory level. It outlines first the spatially varying quantities they predict and the physical principles they use. It then describes the circulations the models simulate and some of the reasons why these circulations are important in the Earth system. Section 3 describes the models at an intermediate level, outlining their governing equations, some approximations used to improve their efficiency and the grids they typically employ. Section 4 outlines at a more technical level the main approximations the models typically use and the steps in the discretisation of their equations, drawing attention to some of the difficulties which limit their accuracy or reliability. Section 5 discusses wider and future perspectives.

Chassignet et al. (2019) provides an alternative non-technical introduction to ocean modelling. McWilliams (1996) and Fox-Kemper et al. (2019) provide more detailed reviews and Griffies (2004) is still a helpful primer on the basic techniques. Aspects of the design, testing, documentation and support for an ocean model code that are crucial for it to be suitable for use in operational predictions or climate simulations are covered in Wan et al. (2024). Porter et al. (2024) discuss the adaptations of ocean models required for them to perform efficiently on modern high-performance computers (HPCs).

## 2. An overview of the models and what they simulate

### 2.1 The quantities simulated and the principles used

The temperature structure of the ocean at a given time in a physical ocean model is represented by a three-dimensional (3D) grid of temperature values. The three dimensions of the grid correspond to the three dimensions of space. One of the dimensions is aligned with the local vertical and the other two with locally horizontal directions. The set of temperature values on the grid is referred to as the temperature field. The salinity structure is similarly represented by a 3D grid of salinity values, referred to as the salinity field. The currents in the two locally horizontal directions are represented by two fields and the locally vertical current by a third field. The fluid's density and pressure are also represented by fields. In total, conceptually there are seven 3D fields (the temperature, salinity, density, pressure and 3 velocity fields) and the physical ocean model simulates how these fields will evolve in time. Given all these fields at time $t$, the model predicts how they will all evolve over the next few minutes or hours, that is over a time-step $\Delta t$, and hence their values at time $t + \Delta t$. Model predictions to days, months or years ahead are generated by performing a large number of time-steps.

The equations used by physical ocean models are based on the physical principles of:

- conservation of momentum (Newton's laws of motion) for each direction in space;
- conservation of the mass of water and salt;
- conservation of energy (the first law of thermodynamics);
- the thermodynamics determining the density at a point from the temperature, salinity and pressure (the equation of state).

Together with information about the momentum, heat and fresh-water exchanged with the atmosphere and sea-ice at the ocean surface and with the solid earth at the bottom of the ocean (the boundary conditions), these 7 sets of constraints are sufficient to determine how the 7 fields will evolve from given initial values at every point of the 7 fields (the initial conditions). In practice, the details of how the equations are used to provide computationally efficient, stable and accurate solutions are quite intricate. The accuracy of the model predictions is primarily limited by the representation of the ocean structure by the values on a grid whose resolution is limited by computational power. Motions at scales comparable to or smaller than the grid are not resolved. The effects of these sub-grid scale (SGS) motions on the resolved scales are calculated by parametrisation schemes. Although these are based on physical principles and detailed studies, their accuracy and reliability are inevitably limited. This is one of the main areas where further research has potential to improve the model simulations.

## 2.2 The circulations simulated and their impacts

The circulations and physical phenomena that these ocean models are typically used to simulate are principally the:

- near-surface boundary layer where there is strong turbulent mixing driven by surface winds and heating or cooling (Large et al. 1994);

- gyre circulations associated with the region, called the thermocline, where the vertical density gradient is strongest. Large-scale displacements in the thermocline are primarily driven by Ekman pumping: in the sub-tropical gyres, the thermocline is bowl-shaped; in the sub-polar gyres it is dome-shaped (chapter 20 of Vallis 2017);
- meridional overturning circulations (MOCs) associated with heat loss and stirring of mixed layers at high latitudes and wind driven upwelling and heat uptake in the Southern Ocean and near the equator (Srokosz et al. 2021);
- western boundary currents (WBCs); the depth mean WBCs are associated with the wind-driven gyre circulations (Pedlosky 1982, chapter 5) and oppositely directed surface and deep WBCs (Hogg 2001) with MOCs;
- mesoscale circulations (with horizontal scales < 100 km) associated with instabilities of the boundary currents and gyre circulations (Robinson 1983)
- sub-mesoscale motions (with horizontal scales < 10 km) that are strongest in the near-surface boundary layer (Taylor & Thompson 2023).

These circulations and phenomena play important roles in the Earth system. For example: the western boundary currents are responsible for very large meridional transports of heat and geographically varying air-sea fluxes which contribute to the shape of atmospheric circulations; interannual variations in the slope of the thermocline along the equator in the Pacific Ocean are an essential component of the El Nino / Southern Ocean (ENSO) phenomenon; the advection of heat by large-scale ocean currents towards ice shelves has a significant impact on their heat balance and evolution (Stewart et al. 2018); and biogeochemical cycles are typically sensitive to the vertical advection of nutrients (Williams and Follows 2011) .

The ocean models can be configured as a component of a coupled system, with models of other components such as the atmosphere, sea-ice, surface waves or biogeochemistry, or as a stand-alone system with suitable data sets providing surface forcing. They can be configured to cover the entire global ocean, or to cover just a limited region with lateral boundary conditions (that are often taken from a model of a larger region). Their initial conditions can be specified by climatologies based on historical measurements or regularly updated by assimilating the latest measurements as in operational forecast systems (Martin et al. 2024). The model coupling, domain, resolution and initial conditions should be chosen to suit the purpose of the modelling and are constrained by the computational resources available.

## 3. A simple description of ocean models

### 3.1 Governing equations

There are many good books on the basics of fluid dynamics. Fluid dynamics is usually formulated using the concepts of vector calculus. Appendix A gives a brief introduction to vector calculus and its application to fluid dynamics, including simplified derivations of Eqs. (1) – (3) below.

Tracers are defined to be properties that fluid parcels retain unchanged with time. Using $\mathcal{T}$ to denote a tracer, $\boldsymbol{u}$ the velocity field and $D/Dt$ the Lagrangian time derivative (following the motion)

$$DT/Dt = \partial T/\partial t + \boldsymbol{u}.\nabla T = 0. \tag{1}$$

The fraction of the mass of water in a fluid parcel due to saline components, $S$, is a tracer and evolves according to the prognostic Eq. (1). Conservation of mass requires that the rate of decrease of mass inside an infinitesimal volume be equal to the fluxes out of its faces and hence that the density, $\rho$, satisfies

$$\frac{\partial \rho}{\partial t} + \nabla.(\rho\boldsymbol{u}) = 0. \tag{2}$$

Combining Eqs. (1) and (2) one obtains an alternative flux form for the evolution of tracers.

$$\frac{\partial(\rho T)}{\partial t} + \nabla.(\rho\boldsymbol{u}T) = 0. \tag{3}$$

The thermodynamics of sea water is rather complex. Vallis (2017) sections 1.5-1.7 give a helpful introduction to it. The macroscopic motions models represent are taken to be in thermodynamic equilibrium and reversible (e.g. not to include mixing). The internal energy of a fluid parcel (following its motion) is then only changed by the heat ($Q$) input into it and the work done on it by pressure forces on it reducing its volume (work done equals force times distance travelled). A potential temperature, $\theta$, can be defined that is equal to the temperature the fluid parcel would have if reversibly moved without input of heat (adiabatically) to a reference height (such as the surface or 2000 m). The potential temperature evolves according to

$$c_p \frac{D\theta}{Dt} = \frac{\theta}{T} Q \tag{4}$$

where $c_p$ is the heat capacity of the sea-water at constant pressure and $T$ is temperature. Ocean models generally use $\theta$ as a prognostic variable. This requires that $T$ and $\rho$ be calculated from the pressure, $p$, $\theta$ and $S$ using the equation of state for sea-water.

The acceleration of fluid particles is determined from Newton's second law of motion: $\boldsymbol{F} = m\boldsymbol{a}_I$. The acceleration $\boldsymbol{a}_I$ in an inertial frame of reference must take into account that the Earth is rotating and that the fluid velocity $\boldsymbol{u}$ is the velocity relative to this rotating frame of motion. Representing the rotation by the vector $\boldsymbol{\Omega}$ which is aligned with the axis of rotation and equal to the rate of rotation, Vallis (2017) section 2.1 nicely shows that

$$\boldsymbol{a}_I = \frac{D\boldsymbol{u}}{Dt} + 2\boldsymbol{\Omega} \times \boldsymbol{u} + \boldsymbol{\Omega} \times (\boldsymbol{\Omega} \times \mathbf{r}). \tag{5}$$

A perfect fluid does not resist shearing motions (Batchelor 1969). Then the force exerted on an infinitesimal element of the surface area of a fluid parcel by the fluid outside is inward and in the direction normal to the surface. So this force $\boldsymbol{F} = -p\hat{\boldsymbol{n}}$, where $\hat{\boldsymbol{n}}$ is the outward pointing normal vector of unit length and by an argument similar to that in Eq. (A.7) one finds that the pressure force on a volume $\delta V$ is given by $-\delta V \nabla p$. The force due to gravity on this cell is downward and equal to its mass $\rho\delta V$ times g. Putting these expressions together for a perfect fluid we infer that

$$\rho\left[\frac{D\boldsymbol{u}}{Dt} + 2\boldsymbol{\Omega} \times \boldsymbol{u} + \boldsymbol{\Omega} \times (\boldsymbol{\Omega} \times \mathbf{r})\right] = -\nabla p - \rho g \hat{\boldsymbol{k}} \tag{6}$$

where $\hat{\boldsymbol{k}}$ is the local unit vector pointing upward.

In fluids, energy input at one scale does not stay at that scale, some "propagates" to larger scales and some to smaller scales. The smaller scales are visible in tracer fields where one sees tongues of tracers drawn out into filaments that become interleaved. The cascade of energy to small scales results in dissipation of energy and vorticity. In the oceans most mixing

occurs on isopycnal (constant density) surfaces. Models are formulated to mix tracers preferentially along isopycnal surfaces (Redi 1982) and aim to constrain the diapycnal mixing to realistic levels. The mesoscale motions in the boundary currents usually derive their energy by extracting potential energy from the sloping isopycnals associated with the currents. Models which only partially resolve mesoscale motions usually include formulations for additional velocities which flatten these sloping isopycnals (Gent & McWilliams 1990). The momentum equations also include terms which drain kinetic energy. These are usually designed to be strongly scale-selective (e.g. biharmonic) in order to drain energy preferentially from the grid-scale. It is important to restrict the grid-scale velocities to levels that do not result in excessive diapycnal mixing of tracers (Ilicak et al. 2012).

## 3.2 Principles of efficiency, accuracy and stability

Ocean models should be designed to accurately represent the motions of interest and to be as efficient in their calculations as possible. It is also highly desirable that they possess analogues of important conservation properties, such as conservation of energy and momentum, and that they have operators that mimic the properties of div, grad and curl for some of the fields.

It is also essential that the model integrations are stable. The prognostic equations are of the form $\partial P / \partial t = R$. When calculating $P$ at timestep $t_n + 1$ nearly all the terms in $R$ need to be written in terms of quantities at step $t_n$ or earlier steps such as $t_n - 1$. If the timestep is too large one of these terms will cause exponential growth of near gridscale fluctuations in $P$. The CFL criterion which requires $c\Delta t < \Delta x$, where $c$ is a speed (such as $|\boldsymbol{u}|$ or the phase speed of a gravity wave), $\Delta t$ is the timestep and $\Delta x$ is the grid-spacing, is of this form (Durran 1999). If the terms in $R$ that are directly related to $P$ are specified using $P$ at timestep $t_n + 1$, a resulting formulation whose timestep is not restricted can usually be found. Such implicit schemes usually require solution of a matrix equation. If the matrix involves the whole 2D or 3D domain its solution is usually costly. Vertical mixing is a fast process (mixing across many grid cells typically happens in one timestep) and implicit schemes result in 1D tridiagonal matrix equations that can be solved robustly and efficiently, so most ocean models use implicit schemes for vertical mixing.

## 3.3 Approximations that improve efficiency

Sound waves in the ocean travel at about 1500 m/s and sea-level variations associated with depth independent motions travel at about 200 m/s. Other motions associated with internal waves (gravity, Kelvin & Rossby waves) and the currents themselves propagate signals at no more than about 3 m/s. Ocean models usually employ approximations that make their solution more efficient by eliminating sound waves and enabling special treatment of the depth independent motions. The Boussinesq approximation takes the ocean density to be treated as a constant except in the gravitational force $-\rho g \widehat{\boldsymbol{k}}$. The conservation of mass (2) then reduces to $\nabla . \boldsymbol{u} = 0$ which says that the fluid is incompressible and the evolution of tracers simplifies to $\partial \mathcal{T} / \partial t +$

$\nabla.(\boldsymbol{u}\mathcal{T}) = 0$. The deliberate omission of $\partial\rho/\partial t$ from (2) eliminates sound waves from the model's solutions. The external mode which is almost depth independent is usually calculated separately as a depth independent mode. It is usually calculated using variables that depend only on the "horizontal" coordinates using time steps that are about 60 times smaller than those used for the 3D calculations.

Another approximation that is commonly used is to neglect the vertical velocities in the vertical component of the momentum

equation. This hydrostatic approximation is valid for motions with horizontal scales that are much larger than their vertical scales. The vertical pressure gradient is then diagnostic (rather than prognostic) and typically satisfies $\partial p/\partial z = -\rho g$.

## 3.4 Model grid cells

Finite difference schemes take cell values to be point values and calculate derivatives explicitly. The advection of tracers might be calculated using (1). Finite volume schemes calculate the fluxes and forces across cell faces and treat cell values as grid cell means. They conserve volume, heat and momentum and usually aim to conserve energy. Most ocean models are formulated using finite volume schemes at least for tracers.

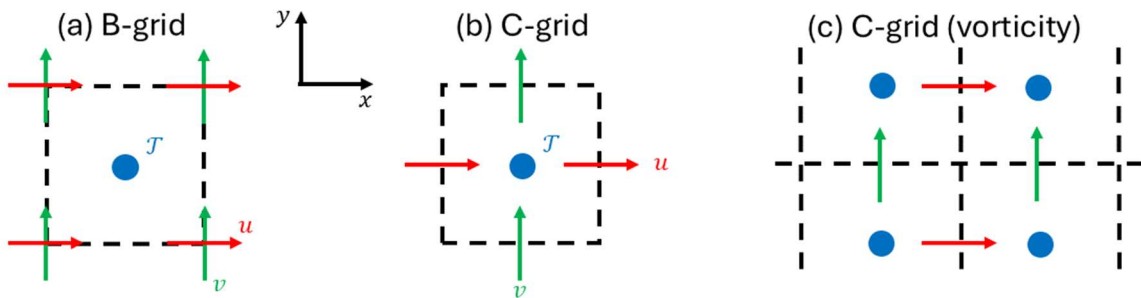

FIG. 1. The horizontal placement of variables on (a) the B-grid and (b) the C-grid. Tracers, $\mathcal{T}$, and velocities $u$ and $v$ in the $x$ and $y$ directions are located at the points marked by blue dots and red and green arrows respectively. (c) shows that on the C-grid the vorticity is naturally centred at the corners of the tracer grid.

Most ocean models use curvilinear orthogonal coordinates in the "horizontal" (on spheroidal surfaces) but an increasing number use triangular or hexagonal grids (Ringler 2010, Korn et al. 2022). Panels (a) and (b) of Fig. 1 illustrate the two most common choices for the placement of variables in grid cells, the Arakawa B and C grids respectively (Arakawa 1960). Both grids store the tracers and the pressure at the centre of each cell. The B-grid stores both components of the velocities at each

of the corners of the cell, whilst the C-grid (Fig. 1(b)) stores them at the centres of the faces to which they are normal and hence at different points. Particularly when the Boussinesq approximation is made, the C-grid is ideal for the evolution of

tracers, conservation of volume and the calculation of $\partial p/\partial x$ at the $u$-points and $\partial p/\partial y$ at the $v$-points. The B-grid is ideal for the calculation of the Coriolis terms, whereas the simplest expression for $v$ at the $u$-point on the C-grid involves a 4-point average of $v$ at the surrounding gridpoints. On the B-grid the horizontal divergence and vorticity are naturally centred at the tracer points, whilst on the C-grid they are centred at the tracer points and the cell corners respectively (Fig. 1(c)).

The choice of vertical coordinate is particularly important in an ocean model. A model level may have a constant height (z-coordinates), have constant potential density (isopycnal coordinates) or vary in proportion to the local depth (terrain-following coordinates). In principle the vertical coordinate could aim to transition from z-coordinates near the sea surface to isopycnal coordinates in the interior and terrain coordinates near the bottom. These coordinates are discussed further in the next section. We note that the axes used by the momentum equations are not altered by these schemes. It is just the coordinates not the axes that are transformed.

Most of the terms in ocean models, including the boundary conditions are only calculated to second order accuracy. This means that if the model were used to simulate an idealised case in which the motions are reasonably well resolved, the errors in the solution should reduce by a factor of 4 as the grid spacing is reduced by a factor of 2. To second order accuracy, a grid cell mean value is equal to the point value at its centre. So in some models it is not entirely clear what the grid cell values are intended to represent. It has been found to be advantageous to calculate the advection terms (usually the fluxes through the cell faces) to higher order accuracy and to limit the values of the fluxes to avoid extending the range of tracer values (Durran 1999, Fox-Kemper 2019). Higher order schemes for the calculation of pressure forces are also advantageous for terrain-following coordinates.

## 4 Methods and approximations employed in ocean models

### 4.1 Variables and equations used

The ocean models used in physical ocean prediction systems evolve 3D fields of the active tracers and the three components of velocity (see section 5.5.1. of Alvarez-Fanjul et al. 2022). They also evolve either a 2D surface pressure (or surface height) field or a 3D pressure field. The active tracers used depend on the formulation of the equation of state. When it is EOS80 (Fofonoff and Millard 1983) the active tracers are potential temperature and practical salinity, whilst when it is TEOS10 (IOC et al. 2010) they are conservative temperature and absolute salinity. The evolution of these fields is determined by some form of the so-called primitive equations (Griffies and Adcroft 2008). The approximations that are usually made are generally well-described in section 5.4 of Alvarez-Fanjul et al. (2022). We note however that the centripetal acceleration is not included in the equations because they have been transformed so that the spheroid coincident with the Earth's bulge, follows a spherical surface (Vallis 2017). It is of course assumed (the turbulent closure hypothesis) that the effect of small-scale motions on large-scale motions can be represented (that is parametrised) in terms of the large-scale motions. None of the Boussinesq, hydrostatic, incompressible and additional Coriolis term approximations is mandatory but maintaining consistent, well-behaved, equations requires care. Some alternative forms of the primitive equations which enjoy good conservation properties are derived in White

et al. (2005). Compressible equations support rapidly traveling sound waves which (can be artificially slowed but) make
competitively efficient solution difficult.

## 4.2 Numerical discretization

Ocean models normally use a smoothly varying horizontal grid consisting of triangular or quadrilateral elements (section 5.4.2. of Alvarez-Fanjul et al., 2022). Where the grid-lines on the quadrilateral grids intersect, they are usually orthogonal (hence called curvilinear orthogonal). The grids are chosen to have rather uniform resolution (cubed sphere grid, Ronchi et al., 1996) or to be isotropic (same resolution locally in the two directions) with grid-spacing decreasing away from the equator and the poles of the grid placed over land (Madec and Imbard, 1996). Triangular elements have the obvious advantage that they can be chosen to follow coastlines more accurately. With triangular elements, reduced grid-spacing is often employed for selected regions within one smoothly varying grid. With quadrilateral elements, reduced grid-spacing is usually achieved by using separate "child" grids that are nested within the "parent" grid with 1-way nesting (the "child" takes boundary values from the "parent" - Staniforth, 1997) or 2-way nesting (the "parent" also takes values from the "child" - Debreu and Blayo, 2008).

Finite difference and finite volume methods are usually employed with the quadrilateral grids. Finite volume models evolve their fields by calculating the fluxes across their cell faces (the difference between the two is not significant for terms that are calculated only to second order accuracy). Models using triangular elements, use either finite element or finite volume techniques (Danilov, 2010; FESOM has transitioned from finite element to finite volume).

The main choices for the staggering of variables on orthogonal grids are the B-grid and C-grid (Arakawa 1960). The dispersion properties of internal waves on the C-grid are better (worse) than the B-grid when the grid resolves (does not resolve) the Rossby radius. Stationary chequer-board modes for the pressure field on the B-grid and the velocity field on the C-grid can be associated with undesirable grid-scale "noise". The dispersion properties of internal waves on triangular grids are more problematic though some finite element pairs (Le Roux et al., 1998) perform relatively well. There has been significant recent progress in the development of C-grid-like formulations for triangular grids (and their hexagonal dual grids) with good, mimetic, properties (Ringler et al., 2010; Cotter and Shipton, 2012).

The choice of vertical "grid" is well known to have far-reaching consequences for ocean models. The Lorenz grid staggering is commonly used despite its computational mode and susceptibility to spurious short-wave instabilities (Arakawa and Moorthi 1988, Bell and White 2017). Ideally, the vertical grid would have fine vertical spacing near the surface, so that the mixed layer can be well represented, and the surfaces on which the vertical coordinate take constant values, would follow isopycnals at mid-depths, so that advective velocities and spurious (numerical) time-mean advective diapycnal transports are minimized, and would follow the bathymetry at the ocean bottom, so that flow down slopes (with its associated vortex stretching) is well represented. Techniques to use coordinates that treat some parts of the motions using Eulerian methods and others using Lagrangian approaches with re-mapping are described in Petersen et al. (2015), Griffies at al. (2020) and Hofmeister et al. (2010). Generation of an appropriate vertical grid for ocean models is an active area of research.

Most terms in ocean models are calculated using second-order accurate formulae. The advection of tracers should however be calculated using schemes of higher order accuracy (typically third or fourth order) which also take care to retain the upper and lower bounds of the advected quantities. There is a very extensive literature on this subject (Durran, 1999, Brasseur and Jacob, 2017) and it is generally agreed that the advecting velocity field should be constrained to be sufficiently smooth (e.g., Ilicak et al., 2012). The effective resolution of the model also depends on how scale-selective the dissipation of variance is near the grid scale (Soufflet et al. 2016).

Specific terms in the equations of motion present different challenges depending on the grid that has been chosen. For terrain-following coordinates, calculation of the horizontal pressure gradient to higher order (Shchepetkin and McWilliams, 2003) and of the diffusion along isopycnal surfaces (Lemarié et al., 2011) is beneficial, and some smoothing of the bathymetry is necessary. Formulation of the governing equations for the cells that are only partially filled by water is an active area of research (Adcroft, 2013; Debreu et al., 2020). For C-grid models, calculation of the Coriolis term should ensure conservation of energy and some care is needed to avoid unintended transfer of energy to the grid-scale (Hollingsworth et al. 1983, Bell et al. 2017, Ducousso et al., 2017).

The strengths and weaknesses of various time-stepping schemes used in ocean models are reviewed in Lemarié et al. (2015). Various approaches have been taken to the time-stepping of the external (barotropic) mode (Shchepetkin and McWilliams, 2003; Demange et al., 2019).

### 4.3 Parameterization of unresolved processes

The parameterization of unresolved processes is of primary importance: Fox-Kemper et al. (2019) provides a useful review. The classic parameterizations of isopycnal diffusion (Redi, 1982; Visbeck et al., 1997), and of the slumping of isotherms by baroclinic instabilities (Gent and McWilliams, 1990) work well in climate models with order $1^\circ$ grid spacing. The latter needs to be developed further for models of higher resolution using ideas such as Bachman (2017) and Mak et al. (2018). It is increasingly clear that sub-mesoscale motions within the ocean surface boundary layer flux heat vertically (Fox-Kemper et al., 2011) and generate filamentary structure. The interaction of these motions with standard parametrisations of turbulence (Umlauf and Burchard, 2005) and Langmuir turbulence (Reichl et al., 2016) is an active area of research as is the parameterization of internal dissipation by internal gravity waves generated by tidal displacements over steep bathymetry (de Lavergne et al., 2020). Machine learning (ML) methods are being applied to the parametrisation of sub-gridscale motions (Zanna & Bolton 2020, Ross et al. 2023) and are likely to play important roles in future ocean models.

### 5. Wider and future perspectives

Modern ocean models use large HPC resources and open source codes supported by communities of scientists and software engineers. They support public safety and protection of the environment by contributing to short-range weather predictions (including forecasts of hurricanes), seasonal forecasts of El Niño and information about climate change. In order to properly

appreciate their roles one needs to see them as one component within the much wider range of scientific activities required to provide this support. Innovations in remote sensing and in situ measurement technology and their internationally-coordinated and sustainable implementation are fundamental to these endeavours. The development of seasonal predictions in the late 1980s and early 1990s, for example, was closely tied to the development of the TOGA TAO array (Smith 2001). The doubling of the number of transistors in a CPU every 2 years from 1970 - 2020 (Porter et al. 2024), and the emergence of accurate near real-time satellite altimetry and the ARGO system of drifters around the turn of the century enabled near global assimilation and prediction of the strongest mesoscale ocean motions to first become a reality around 2015 (Bell et al. 2015). What will be the major societal drivers and what are the best opportunities for scientific improvement in the next 10-20 years? We don't have a crystal ball but we can offer some suggestions.

As mentioned at the end of the last section, ML methods have recently emerged as a new set of tools that can be used in many ways to improve Earth system models (Eyring et al. 2024). Depending on the directions explored, the ocean model codes may need to be rewritten as differentiable functions to exploit ML methods fully (Silvestri et al. 2024). Ocean reanalyses based on measurements from 1980 onwards are gradually being improved and together with atmospheric reanalyses will provide an essential resource for inputs to ML and the assessment and improvement of coupled ocean-atmosphere models. The international coordination established under CMIP (Coupled Model Intercomparison Project, https://www.wcrp-climate.org/wgcm-cmip) should enable much richer sets of experiments to be conducted and more diverse ensembles of ocean and Earth system models to be explored than would otherwise be possible. There is also scope for more traditional improvements to ocean models; such as improved methodologies and choices for: vertical coordinates; parametrisation of vertical mixing; specification of surface exchanges (Yu 2019, Storto et al. 2024); the use of finer horizontal resolution in selected regions; and more efficient generation of ensembles of simulations. Coupled simulations of ENSO still have significant deficiencies and simulations of the future Atlantic MOC are not as reliable as they need to be. In summary, it is reasonable to be optimistic that successful progress with significant societal impacts can be made over the next 10-20 years.

**Appendix A An introduction to vector calculus for fluid dynamics**

Fluid dynamics is concerned with properties like temperature and salinity that vary spatially and evolve with time. Such properties are referred to as fields. Just as $y(x)$ represents any curve $y$ that is a function of $x$ in ordinary calculus, $F(x, y, z, t)$ represents any field that depends on $x, y, z$ and $t$. In ordinary calculus we have $\delta y \cong y(x + \delta x) - y(x)$ and consider $\delta y / \delta x$ in the limit as $\delta x$ becomes very small. For "smooth" enough functions there is a limiting value $dy/dx$. In vector calculus we consider how $F$ varies with each of its coordinates whilst keeping the other coordinates fixed. Varying $x$ and considering the limit when $\delta x$ becomes very small we write

$$\frac{\partial F}{\partial x} = \frac{\partial F}{\partial x}\bigg|_{y,z,t} = \frac{F(x+\delta x,y,z,t)-F(x,y,z,t)}{\delta x} \quad \text{in the limit as } \delta x \to 0. \tag{A.1}$$

$\partial F / \partial x$ is termed the partial derivative of $F$ with respect to $x$. The variables that are held constant can be explicitly declared as shown. For brevity they are often omitted, in which case they are implicit. An extremely useful expression analogous to $\delta y \cong y(x + \delta x) - y(x)$ is

$$\delta F \cong \frac{\partial F}{\partial x}\delta x + \frac{\partial F}{\partial y}\delta y + \frac{\partial F}{\partial z}\delta z + \frac{\partial F}{\partial t}\delta t. \tag{A.2}$$

(a) A curve in 3D space

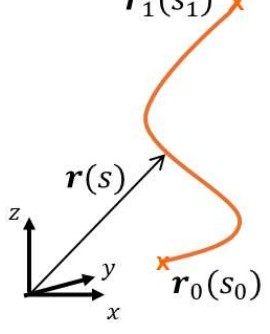

(b) A contribution to $\boldsymbol{V}.\boldsymbol{U}$

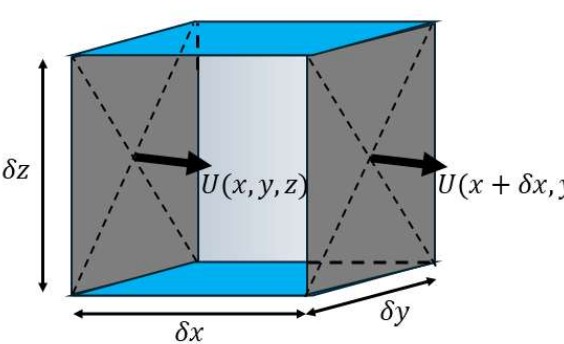

(c) The path used to compute the vorticity

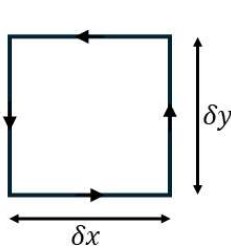

**FIG A1**: (a) Illustration of a curve $\boldsymbol{r}(s)$ in 3D space obtained by varying the scalar parameter $s$ from $s_0$ to $s_1$. (b) Illustration of the contribution to the mass flux divergence for a cell of volume $\delta x \delta y \delta z$ from the fluxes through the faces perpendicular to the $x$-axis. (c) The anti-clockwise path around the sides of the infinitesimal cell with sides of length $\delta x$ and $\delta y$ used to calculate the area integral within the cell of the normal component of vorticity.

For the sake of simplicity we limit ourselves hereafter to rectilinear Cartesian coordinates in which the axes are orthogonal straight lines, the coordinates of a point $\boldsymbol{r}$ are denoted by $(x, y, z)$, the distance from the origin, $d$, is given by Pythagoras' theorem ($d^2 = x^2 + y^2 + z^2$), and $z$ points upward. We explain later that the equations can be derived for a more general set of locally orthogonal coordinates.

Consider first a curve $\boldsymbol{r(s)}$ between two points, $\boldsymbol{r}_0 = \boldsymbol{r}(s_0)$ and $\boldsymbol{r}_1 = \boldsymbol{r}(s_1)$ as illustrated in Fig. A1(a). Integrating Eq. (A.2) along the curve (with $\delta t = 0$) one sees that

$$F(\boldsymbol{r}_1) - F(\boldsymbol{r}_0) = \int_{s_0}^{s_1} \left( \frac{\partial F}{\partial x}\frac{dx}{ds} + \frac{\partial F}{\partial y}\frac{dy}{ds} + \frac{\partial F}{\partial z}\frac{dz}{ds} \right) ds \tag{A.3}$$

Writing $\nabla F = (\partial F / \partial x, \partial F / \partial y, \partial F / \partial z)$ and $d\boldsymbol{r}/ds = (dx/ds, dy/ds, dz/ds)$, (3) can be re-expressed as

$$F(\boldsymbol{r}_1) - F(\boldsymbol{r}_0) = \int_{s_0}^{s_1} \nabla F . \frac{d\boldsymbol{r}}{ds} ds = \int_{r_0}^{r_1} \nabla F . d\boldsymbol{r} \tag{A.4}$$

Eq. (A.4) is the defining property of $\nabla F$ which is termed the gradient of $F$ or grad $F$ for short. If one integrates around any path which closes on itself, i.e. $\boldsymbol{r}_1 = \boldsymbol{r}_0$, one sees that the left-hand side of Eq. (A.4) is equal to zero. Hence the integral of $\nabla F$ around any closed curve is zero.

The rate of change with time of a field $F$ following a fluid particle moving at velocity $\boldsymbol{u} = (u, v, w)$ can also be inferred from Eq. (A.2) by dividing it by $\delta t$. Following the fluid parcel, $\delta x \cong u\delta t$, $\delta y \cong v\delta t$ and $\delta z \cong w\delta t$. So

$$\frac{DF}{Dt} = \frac{\partial F}{\partial t} + u\frac{\partial F}{\partial x} + v\frac{\partial F}{\partial y} + w\frac{\partial F}{\partial z} = \frac{\partial F}{\partial t} + \boldsymbol{u}.\nabla F. \tag{A.5}$$

Here we have used the standard notation $DF/Dt$ to denote the rate of change of $F$ with respect to time following a fluid parcel, which is often called the Lagrangian derivative. Tracers are defined to be properties that fluid parcels retain unchanged with time. Using $\mathcal{T}$ to denote a tracer we see that

$$D\mathcal{T}/Dt = \frac{\partial \mathcal{T}}{\partial t} + u\frac{\partial \mathcal{T}}{\partial x} + v\frac{\partial \mathcal{T}}{\partial y} + w\frac{\partial \mathcal{T}}{\partial z} = 0. \tag{A.6}$$

An equation expressing conservation of mass can be derived by considering the "notional" cuboid cell illustrated in Fig A1(b). The density of a fluid, $\rho$, is defined to be its mass per unit volume. The volume of the cell in Fig. A1(b) equals $\delta\mathcal{V} = \delta x\delta y\delta z$. The fluxes of mass through the two side-faces perpendicular to the $x$-axis are $U(x, y, z)\,\delta y\delta z$ and $U(x + \delta x, y, z)\,\delta y\delta z$ where $U = \rho u$. So in the limit as the cell volume becomes very small the flux out of the cell from these two faces equals

$$[U(x + \delta x, y, z) - U(x, y, z)]\delta y\delta z \cong \frac{\partial U}{\partial x}\delta x\delta y\delta z. \tag{A.7}$$

Conservation of mass requires that the increase in mass inside the cuboid plus the fluxes out of the three pairs of side-faces equal zero. Using expressions corresponding to Eq. (A.7) and dividing by $\delta\mathcal{V}$ one obtains

$$\frac{\partial\rho}{\partial t} + \frac{\partial(\rho u)}{\partial x} + \frac{\partial(\rho v)}{\partial y} + \frac{\partial(\rho w)}{\partial z} = \frac{\partial\rho}{\partial t} + \nabla.(\rho\boldsymbol{u}) = 0. \tag{A.8}$$

The operator $\nabla.$ introduced in Eq. (A.8) is called the divergence. At any point it is defined to be the outward flux per unit
volume through a surface enclosing that point. Gauss' theorem shows that for "smooth" fields the divergence does not depend on the shape of the volume (e.g. it is the same for infinitesimal spheres and cuboids). Combining Eqns. (A.6) and (A.8) one obtains the flux form for the conservation of tracers.

$$\frac{\partial(\rho\mathcal{T})}{\partial t} + \frac{\partial(\rho u\mathcal{T})}{\partial x} + \frac{\partial(\rho v\mathcal{T})}{\partial y} + \frac{\partial(\rho w\mathcal{T})}{\partial z} = \frac{\partial(\rho\mathcal{T})}{\partial t} + \nabla.(\rho\boldsymbol{u}\mathcal{T}) = 0. \tag{A.9}$$

There is one other vector quantity that is particularly important in fluid dynamics: the curl of the velocity field, $\nabla \times \boldsymbol{u}$, which
is termed the vorticity. The component of the vorticity perpendicular to the infinitesimal square shown in Fig. A1(c) is calculated by considering the line integral of $\boldsymbol{u}.d\boldsymbol{r}$ anti-clockwise around its sides. Similarly to Eq. (A.7), $[v(x + \delta x, y) - v(x, y)]\delta y \cong \frac{\partial v}{\partial x}\delta x\delta y$ and

$$\oint \boldsymbol{u}.d\boldsymbol{r} = \iint\left(\frac{\partial v}{\partial x} - \frac{\partial u}{\partial y}\right)\mathrm{d}x\mathrm{d}y = \iint \nabla \times \boldsymbol{u}.d\boldsymbol{S}. \tag{A.10}$$

Here $d\boldsymbol{S}$ is the vector perpendicular to the area enclosed by the line integral whose length is equal to that area. Stokes' theorem
shows that the vorticity does not depend on the shape of the area used to calculate it (e.g. it is the same for infinitesimal circles and squares). The vorticity of the fluid is particularly important because of Kelvin's theorem which states that under certain conditions following a fluid parcel the vorticity does not change with time (i.e. it is conserved). Ertel's theorem on conservation of potential vorticity is based on Kelvin's theorem (Pedlosky 1982 chapter 2).

Expressions for the gradient, divergence and curl of vector fields and relations between them can be derived for generalised
curvilinear orthogonal coordinate systems (see Lorrain and Corson 1970 for a well illustrated introduction and Appendix A of
Batchelor (1967) for a concise summary). Latitude, longitude coordinates for the sphere are one example of such coordinate
systems.

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

**Competing interests**

The contact author has declared that none of the authors has any competing interests.

**Data and/or code availability**

No original code or data were used to produce this paper.

**Authors contribution**

Bell wrote most of the text of all the sections. Ciliberti wrote the first draft of section 2. Schiller reviewed and made improvements to all the sections.

**Acknowledgements**

We gratefully acknowledge advice and insights from many colleagues over many years.

