# Peer review of "Numerical Models for Simulating Ocean Physics"

_State of the Planet, 2024_

## Author Response (AR2)

**Responses to Reviewers and Description of main changes to the paper**

In response to the reviewers' comments we have:

- added a second paragraph to the introduction pointing the reader to other introductions to ocean models;
- introduced a new section (new section 3) which bridges the large gap between the original elementary level section 2 and advanced level section 3 by describing the models at an intermediate level;
- introduced appendix A intended to bridge the gap between section 2 and the new section 3;
- introduced a final section (new section 5) which briefly describe the wider context and future prospects for the development of these models;
- (of course) made a number of additional changes improving various details in the paper.

We have also adjusted the abstract and first paragraph of the introduction so they align with the above revisions.

We have made one additional change in the new section 4 where in lines 247-249 we have included the sentence: "The Lorenz grid staggering is commonly used despite its computational mode and susceptibility to spurious short-wave instabilities (Arakawa and Moorthi 1988, Bell and White 2017)." These short-wave instabilities have recently been found in some simulations of the Mozambique channel (in a paper submitted to JAMES by Claire Menesguen and co-authors) and are not well known so we think it useful to include these references.

**Responses to Reviewer 1**

Thank you for your comments. We have drafted some substantial revisions to the paper based on them – as explained below.

**It is a daunting task to summarize ocean numerical model physics in a few pages.**

We can only agree with you. Our original draft was aimed at people who are responsible for implementing ocean models in operational systems. So it included only the technical section. We were asked to provide some more introductory material and we wrote that in section 2 at an elementary level.

**My comments are as follows:**

1. Contrast between Sections 2 and 3: Section 2 is written at an elementary level with no references while Section 3 is technical with many references. While this is announced in the abstract, the transition is too abrupt. It is almost as if it was written by two different contributors. I suggest that the authors slightly rewrite both sections for a better flow/continuity.

We can see that there is a problem with the lack of continuity. We have considered various options to address that and have made two types of changes. The main change is that we have introduced a new section (3.5 pages long) that describes the models at an intermediate level. This outlines their governing equations, some approximations used to improve their efficiency and the grids they typically employ. We felt that there was still quite a jump between the

elementary and intermediate levels so we have also written a 2-page appendix that explains the vector calculus underlying fluid dynamics for people who are familiar only with the calculus of a single independent variable, i.e. y(x). We don't remember seeing explanations (rather than summaries) of this sort and think it may be useful for people who would otherwise be unable to understand the equations in section 3. It may also be a useful reminder for people who last covered this material many years ago. This is quite a difficult thing to do well so we have run it past one or two people to see if it works for them. The second (relatively minor) change is that we have added suitable references in subsection 2.2. This makes that subsection more informative and slightly less elementary.

 The paper is lacking a summary section. I suggest discussing the current state of the art and where numerical models might be 20 years from now. An example can be found in the introductory encyclopedia paper of Chassignet et al. (2019) (https://www.coaps.fsu.edu/pub/eric/papers\_html/Chassignet\_et\_al\_19.pdf) which also aims at reaching a broad audience.

We have included a new final section (new section 5) titled "Wider and future perspectives" that contains just two paragraphs. The first outlines the role of ocean models as one component within a complex network of activities aiming to support public safety and protection of the environment. It also provides some historical context. The second paragraph discusses prospects for ocean models over the next 10-20 years within this wider context.

**3. As in Chassignet et al. (2019), I suggest adding a "Further Reading" section for the readers that interested in more details.**

After re-reading Chassignet et al (2019), we decided to end the introduction with a paragraph drawing the readers' attention to alternative introductions to ocean modelling. This paragraph begins "Chassignet et al. (2019) provides an alternative non-technical introduction to ocean modelling. McWilliams (1996) and Fox-Kemper et al. (2019) provide more detailed reviews and Griffies (2004) is still a helpful primer on the basic techniques." We think this is the most helpful short-list of introductions we can provide. We cite many books and papers in the main text.

4. Minor comment: In the Introduction, on line 20, you introduce Section 4 of Wan et al. It is a bit awkward as it follows the description of Section 3 of your paper and you do not have a Section
4. I assume Wan et al. is part of the same issue - I suggest rephrasing the sentence to make it clear that it is complementary to this paper.

Yes, Wan et al. is part of the same issue and we can see the sentence needed re-phrasing. The introduction now concludes by saying "Aspects of the design, testing, documentation and support for an ocean model code that are crucial for it to be suitable for use in operational predictions or climate simulations are covered in Wan et al. (2024). Porter et al. (2024) discuss the adaptations of ocean models required for them to perform efficiently on modern high-performance computers (HPCs)." These points explain why those topics are not otherwise mentioned in our review.

**Responses to Reviewer 2**

Thank you for your review. We agree with nearly all your points and have revised a new draft of our paper accordingly as described below. In response to reviewer 1 we have added an additional section (new section 3) which describes ocean models at an "intermediate" level. We have also added some references to section 2.2.

This is a very short, elementary overview of ocean modeling, targeting mainly "largescale" simulations of the ocean circulation. Given the limited space, the manuscript is necessarily subjective, especially in the choice of references, which is fine. Several in-depth review papers (and reference texts) are cited, which is good. I think the manuscript is a useful short introduction. I do have a few comments which should be addressed prior to publication.

Yes it is difficult not to be subjective. The last paragraph of the introduction now draws attention to other introductions to ocean modelling which emphasise different things. The last section of the paper is aimed at expert model users and deliberately focuses on some more advanced technical detail.

**Around line 45:**

**In addition to the 7 equations, it might be useful to also mentioning the surface, bottom, and lateral boundary conditions (the surface being the most relevant one)**

Yes! We should certainly have done that. In the new version after the list of physical principles we have now written "Together with information about the momentum, heat and fresh-water exchanged with the atmosphere and sea-ice at the ocean surface and with the solid earth at the bottom of the ocean (the boundary conditions), these 7 sets of constraints are sufficient to determine how the 7 fields will evolve from given initial values at every point of the 7 fields (the initial conditions)." The paper does not say much about surface fluxes, which is a very large topic in its own right but the final section now refers to Yu (2019) and Storto et al. (2024).

Yu, L.: Global Air–Sea Fluxes of Heat, Fresh Water, and Momentum: Energy Budget Closure and Unanswered Questions. Annu. Rev. Mar. Sci., 11, 227–48, 2019.

Storto, A., Frolov, S., Slivinski, L., and Yang, C.: Correction of Air-Sea Heat Fluxes in the NEMO Ocean General Circulation Model Using Neural Networks, Geosci. Model Dev. Discuss. [preprint], https://doi.org/10.5194/gmd-2024-185, in review, 2024.

**line 45-51:**

**A distinction between "resolution" and "grid" spacing might be appropriate. The two aren't the same but are treated as such. It takes multiple (at least 4) grid spacings to "resolve" a process, such that for a given grid spacing dx, we may begin to resolve a processes of size 4\*dx.**

Yes that is right. We have revised the relevant sentences slightly. They now say: " Motions at scales comparable to or smaller than the grid are not resolved. The effects of these sub-grid scale (SGS) motions on the resolved scales are calculated by parametrisation schemes". Later in the paper, between lines 132 and 133 in the old version, we have added: "The effective resolution of the model also depends on how scale-selective the dissipation of variance is near the grid scale (Soufflet et al. 2016)."

Soufflet Y., Marchesiello P., Lemarie F., Jouanno J., Capet X., Debreu L., and Benshila R.: On effective resolution in ocean models. Ocean Modelling, 98, 36–50, https://doi.org/10.1016/j.ocemod.2015.12.004, 2016.

**line 50:**

**You mention that parameterizations are inevitably limited. Please spend one sentence as to why? (Structural and parametric uncertainty, lack of calibration, discretization errors).**

Well, the fundamental issue with most parametrisations in ocean models is that they are trying to represent the effect of motions that are not resolved. We have already said that. Structural and parametric uncertainty are consequences of that issue. As the discussion here is intended to be at a non-technical level we believe the current text is appropriate.

**line 60 on MOC:**

It is not only convective mixing in high lats, it is also boundary mixing, as revealed in the OSNAP East measurements.

**Furthermore, closure of the global MOC is also through a range of mixing processes.**

Some of the mixing in the boundary currents will be convective mixing. But some of the mixing in the open ocean will be at least partly driven by wind stirring. So on reflection we think it is better to omit the word "convective". We have re-organised the sentence somewhat so it now reads "meridional overturning circulations (MOCs) associated with heat loss and stirring of mixed layers at high latitudes and wind driven upwelling and heat uptake in the Southern Ocean and near the equator (Srokosz et al. 2021);" We believe Srokosz et al 2021 is a good reference as it introduces a series of recent reviews of the MOC.

Srokosz, M., Danabasoglu, G., Patterson, M.: Atlantic Meridional Overturning Circulation: Reviews of observational and modeling advances - An introduction. *Journal of Geophysical Research: Oceans*, **126**, **1**, https://doi.org/10.1029/2020JC016745, 2021.

**line 62 on boundary currents:**

Those have nothing to do with the MOC, i.e., they exist regardless of the MOC. Please remove "MOC" mention here (westward intensification goes back to the models by Stommel (1948) and Munk (1950)).

It was not very clear what we meant in the old version. So we have rewritten this bullet point distinguishing more carefully between depth mean WBCs associated with the wind-driven gyres and vertically varying WBCs that are part of the MOCs: "western boundary currents (WBCs); the depth mean WBCs are associated with the wind-driven gyre circulations (Pedlosky 1982, chapter 5) and oppositely directed surface and deep WBCs (Hogg 2001) with MOCs;"

Hogg, N. G.: Quantification of the deep circulation, 259-270. In Siedler, G., Curch, J., Gould, J. Ocean circulation and climate: observing and modelling the global ocean, International Geophysics Series vol 77, San Diego, Academic Press, 2001.

Pedlosky, J.: Geophysical Fluid Dynamics New York, Springer-Verlag, 624 pp, 1982.

**line 70:**

**I think you mean "mass balance", not "heat balance".**

We could have written mass balance or heat balance here; the advection of warm water affects how much ice melts. We prefer to retain "heat balance".

line 75-78:

You could reference papers here on data assimilation and sea ice modeling that will appear in the same issue as part of OceanPrediction.

Yes. We have included a reference to the companion paper by Martin et al. (2014). We haven't referenced the sea-ice modelling paper.

line 90:

"...the elliptical geoid of the Earth's bulge follows a spherical surface"

This is a bit obscure. The geoid proper deviates from the reference ellipsoid (by about +-80 metres).

I think what you mean is that the centrifugal term is absorbed in the gravitational term by means of a geopotential).

It would be good to clarify your sentence.

Yes this sentence was not entirely clear for the reason you give. The Earth's bulge due to its rotation is 20 km (equatorial radius = 6378 km; polar radius = 6357 km). So the geoid does follow a nearly elliptical surface. We have rewritten the sentence so that it says "... the centripetal acceleration is not included in the equations because they have been transformed so that the spheroid coincident with the Earth's bulge, follows a spherical surface (Vallis 2017).